# Portable DC Supply Based on SiC Power Devices for High-Voltage Marx Generator

**Jacek Rąbkowski \***, **Andrzej Łasica**, **Mariusz Zdanowski**, **Grzegorz Wrona and Jacek Starzyński**

Faculty of Electrical Engineering, Warsaw University of Technology, ul. Koszykowa 75, 00-662 Warsaw, Poland; andrzej.lasica@ee.pw.edu.pl (A.Ł.); mariusz.zdanowski@ee.pw.edu.pl (M.Z.); grzegorz.wrona@ee.pw.edu.pl (G.W.); jacek.starzynski@ee.pw.edu.pl (J.S.)
\* Correspondence: jacek.rabkowski@ee.pw.edu.pl

**Abstract:** The paper describes major issues related to the design of a portable SiC-based DC supply developed for evaluation of a high-voltage Marx generator. This generator is developed to be a part of an electromagnetic cannon providing very high voltage and current pulses aiming at the destruction of electronics equipment in a specific area. The portable DC supply offers a very high voltage gain: input voltage is 24 V, while the generator requires supply voltages up to 50 kV. Thus, the system contains two stages designed on the basis of SiC power devices operating with frequencies up to 100 kHz. At first, the input voltage is boosted up to 400 V by a non-isolated double-boost converter, and then a resonant DC-DC converter with a special transformer elevates the voltage to the required level. In the paper, the main components of the laboratory setup are presented, and experimental results of the DC supply and whole system are also shown.

**Keywords:** Marx generator; high-voltage; SiC; DC-DC converters; DC supply

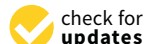



## 1. Introduction

Marx generators are still the most popular systems used to generate high-voltage pulses. In addition to the typical microsecond voltage surges used to test power devices in accordance with the PN-EN 60060-1 standard [1], tests using pulses with a rise time of nanoseconds are becoming more and more common. They are used in industry [2–4], medicine [5] and scientific research, where they are used for electroporation-defunctionalization of cell membranes [6,7], which can be used for sterilization, but also for the penetration of cells and their organelles by chemical compounds (e.g., drugs) or genetic material. However, most applications of this type of pulses are in electromagnetic compatibility [8,9], where they can simulate nuclear electromagnetic pulses (NEMPs) or high-altitude electromagnetic pulses (HEMPs) when testing civil or military equipment, e.g., according to the MIL-STD-461 standard [10,11]. Examples of such generators can be found in the portfolio of different companies, such as in [12]; however, the pulses produced by the generators of the mentioned manufacturer show a rise time of 2.3 ± 0.5 ns and are charged from DC power supplies with voltages of 0.2 kV to 25 kV, with positive polarity only [13]. Tests with the use of this type of generator are often performed outside laboratories, on open training grounds [14]. Hence, it is recommended that the design of the DC power supply should be as light and compact as possible, which will facilitate transport. An additional advantage will be the battery power supply, which enables conducting research even on test sites not equipped with auxiliary infrastructure. Such a system requires portable DC power supplies providing voltages up to 50 kV but with the feed from low-voltage batteries.

Taking into account these requirements, a portable DC power supply has been developed on the basis of silicon carbide (SiC) power device technology. The first step of the research was a literature review in the area of high-voltage power supplies and it

was observed that most solutions use various types of transformer-based DC-DC converters [15–22]. In [15–17], a single active bridge was applied, while in some other works, a resonant converter can be found [19–22]. What is also interesting is a series connection of the DC-DC converters: a parallel-input series-output structure was discussed in [18], while a special topology was developed in [22], and in [19], a voltage multiplier was applied. Most of the solutions are supplied from the voltage in the range of hundreds of volts (i.e., three-phase rectifier) and use single-stage DC-DC conversion to reach the output voltage in a required kV range. All in all, in most cases, traditional silicon power devices were applied for operating, in most cases at tens of kHz. Therefore, the goal of this work was to verify the performance of new SiC power devices. Then, as the voltage of the input batteries is rather low, a two-stage system was considered with an additional non-isolated boost converter. The expected gain of this converter was relatively high (up to 18); thus, several concepts were reviewed starting from the charge pump [23] through to impedance source topologies [24,25]. Finally, the double-boost topology [26] was found to be most suitable; however, SiC devices are considered to reduce the size of the passive components.

## 2. The Setup of the Marx Generator

The main goal of the Marx generator to be supplied is to generate nanosecond high-voltage pulses for exposure tests of electronic equipment. The generator load will be a stripline impedance of 130 $\Omega$ and the expected output voltage from the generator is 1 MV. Therefore, taking into account the available supply systems and 50 kV-rated capacitors, it was decided to build a 20-stage system (Figure 1a). The capacitance of each capacitor is 8 nF, while the predicted repetition time of the generated pulses is 1 Hz. Therefore, in total, the DC power supply must charge the 160 nF capacity to 50 kV in less than 1 s. As charging resistors, volume resistors were used, which are immune to short-term current pulses, temporarily significantly exceeding the rated long-term current of these resistors. The value of each resistor marked as $R_c$ in Figure 1a is 6 k$\Omega$. The generator structure itself was placed in a sealed housing (Figure 2a). During the generator's operation, a pressure of several atmospheres was maintained in the housing, which protected the system against surface discharges at higher charging voltages. Due to the fact that strong electromagnetic disturbances are generated in the vicinity of the measuring setup during the generation of high-voltage pulses, it was necessary to place the DC power supply in a sealed metal housing (Faraday cage), and communication between the user and the power supply needed to be carried out via a fiber optic link. Further, the generator ignition initiating signal was sent from the power supply to the optical fiber triggering system.

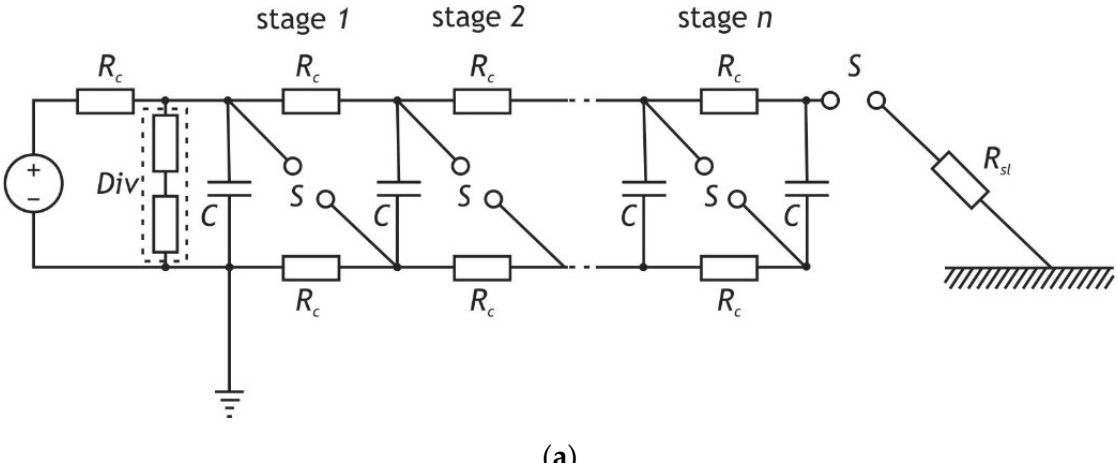

(**a**)

**Figure 1.** *Cont.*

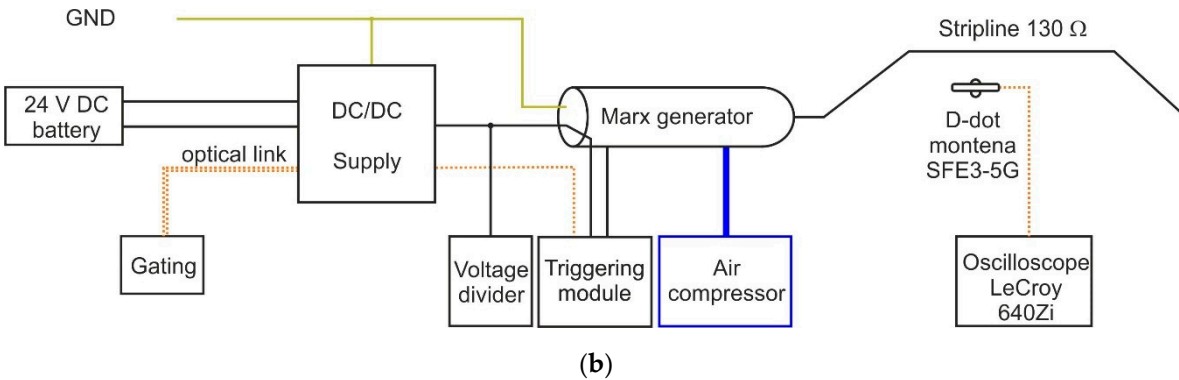

**Figure 1.** Scheme of the Marx generator with 20 stages (**a**) and block scheme of the whole setup (**b**).

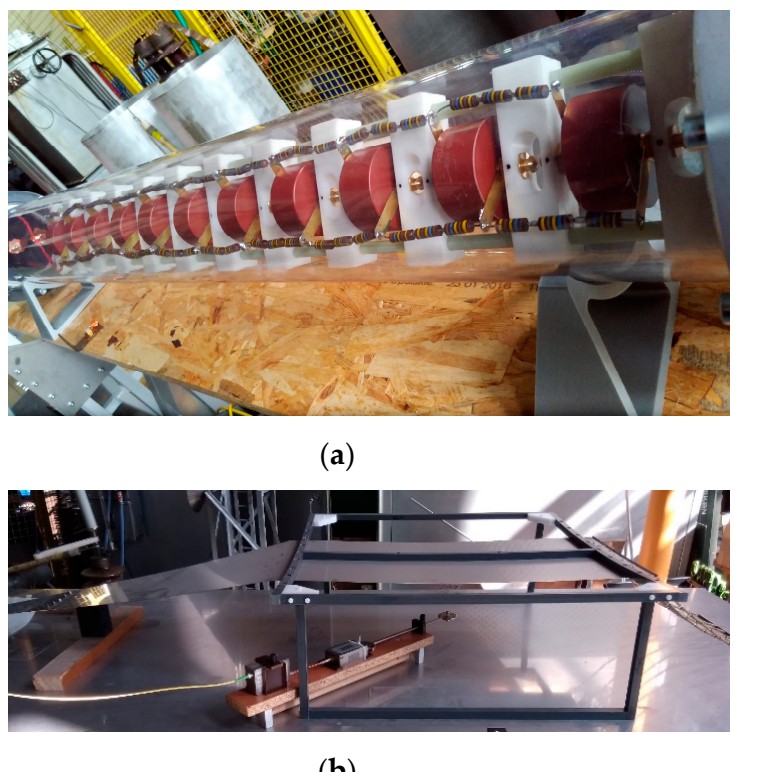

**Figure 2.** Photo of the Marx generator in hermetic sealing (**a**) and stripline with the field sensor (**b**).

Since the voltage measurement of hundreds of kilovolts and rise times of the order of nanoseconds are not feasible with classical voltage dividers, the measurements of the output pulse were measured indirectly—by measuring the field strength on the stripline connected to the generator output (Figure 2b). The dimensions of the line were as follows: length: 1 m, width: 0.48 m, height: 0.3 m, while the total impedance was equal to 130 Ω. A Montena SFE3-5G probe (Montena, Rossens, Switzerland) was used to measure the electric field strength in the stripline space, which can measure pulses with rise times from 110 ps.

## 3. Portable DC Power Supply

The abovementioned requirements for a portable DC power supply are very challenging. At first, the input voltage from the low-voltage battery was assumed to be 24 V, while the nominal output voltage was expected to reach 50 kV when charging the 160 nF capacitance of the Marx generator. This means that the voltage gain of the system exceeds 2000. Moreover, the volume and weight should also be reasonable to make this unit portable. On the other hand, the system was designed to survive electromagnetic impulses of the Max generator placed at a close distance. Finally, control of the charging process is also problematic as precise measurement of the output voltage including signal isolation is difficult.

The proposed solution was based on the two conversion stages presented in Figure 3. The second stage is an isolated series resonant DC-DC converter operating with a fixed voltage gain ($G = 125$) up to 50 kV, while at the input, a non-isolated DC-DC converter is connected. This converter plays the role of a regulated voltage source with the reference voltage proportional to the expected value of the output $V_O$. This approach is more convenient than the complex control of the series resonant DC-DC as precise adjustment of the voltage up to 400 V is less complex. Thus, the reference voltage $V_{DC}$ was set according to the reference $V_O$ and the ideal gain $G$. Then, on the basis of the isolated measurement of the output voltage $V_O$, which was designed to be isolated from the controller and, in consequence, is rather slow, necessary corrections of the $V_{DC}$ were also introduced. This approach was necessary due to variations in the gain of the isolated DC-DC converter with the load changes. Moreover, the voltage of the battery $V_{BAT}$ may also drop with the current. All in all, the proposed solution is simple and easy to manage at high operation frequencies and the output voltage is properly controlled according to the Marx generator's needs.

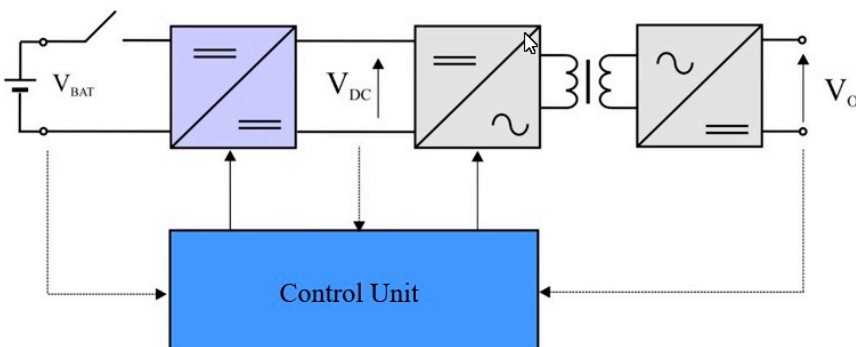

**Figure 3.** Overall scheme of the portable DC power supply.

### 3.1. Double-Boost DC-DC Converter

According to assumed approach, the main task of the input stage is to adjust the voltage levels between the input power circuit (a battery with a constant rated voltage $V_{BAT}$ of 24 V) and the intermediate circuit ($V_{DC}$ up to 400 V). As a result of the analysis carried out earlier, the topology of the double voltage boost was selected as can be seen in Figure 4a. The main circuit of the converter (Figure 4b) consists of two branches, each of which includes one SiC MOSFET (C3M0065090D) and a SiC Schottky diode (C3D16065D) (Cree/Wolfspeed, Durham, NC, USA). The transistors are controlled by integrated gate drivers (ACPL-P343-000E) (Broadcom, San Jose, CA, USA), which also provide isolation between the controller and power circuit. These drivers provided by the separated DC-DC converters (MGJ2D241505SC) (Murata Power Solutions Inc., Westborough, MA, USA) supply the transistor gates with a positive voltage of +15 V to switch them on and −5 V to switch them off. The converters are fed directly from the system input—24 V battery.

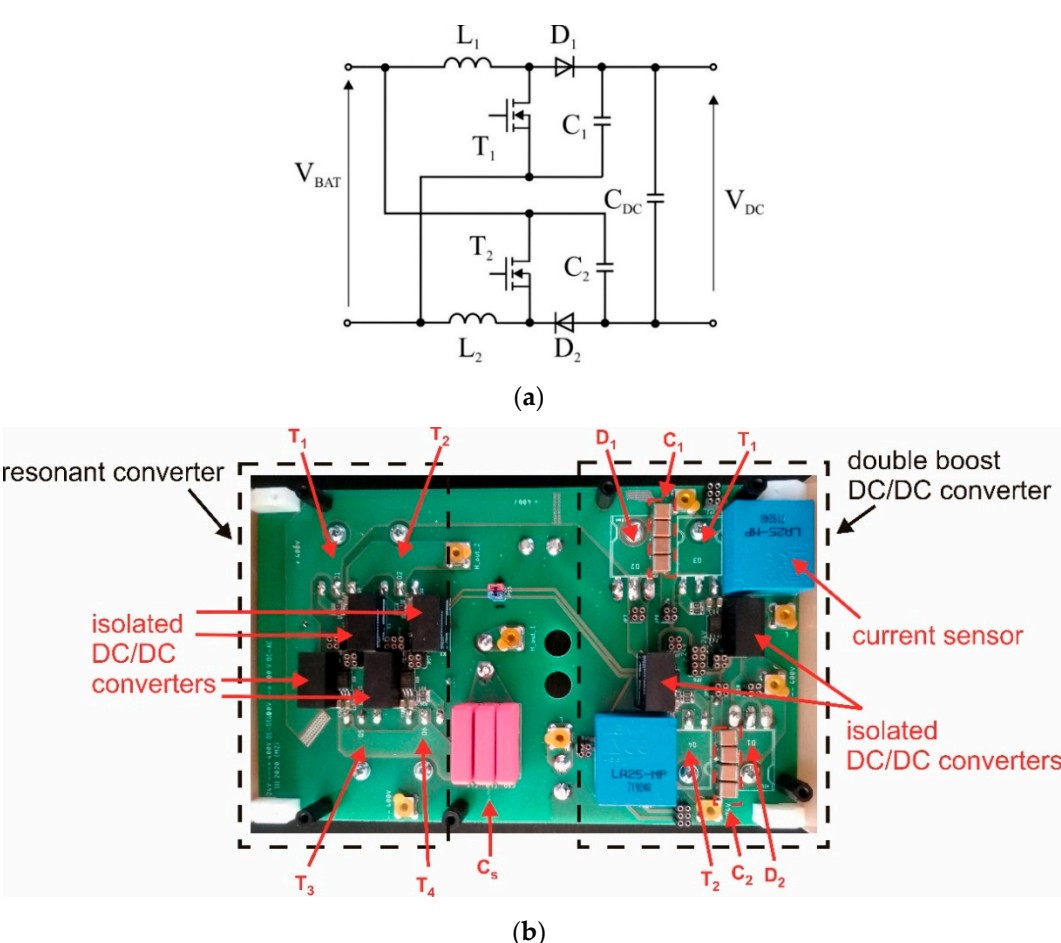

(**a**)

(**b**)

**Figure 4.** Scheme of the double-boost DC-DC converter (**a**) and top view of the common power board with designed prototype (**b**).

To control the input current drawn from the battery, the main circuit was also equipped with two current sensors (LA 25-NP) (LEM USA Inc., Milwaukee, WI, USA), and then the signal was transferred to the control board described in the following sections. Pads were attached to the midpoints of the branches, which enable the connection of the boost inductors of 30 µH/30 A max required in this topology. In parallel, two sets of capacitors, $C_1 = C_2 = 40$ µF/250 V and $4 \times 2.2$ µF/250 V, were applied. As can be seen, the high switching frequency, set to 100 kHz due to the outstanding performance of the SiC devices, enables a reduction in passive components necessary to provide a suitable DC voltage. Moreover, two phase legs operating with a phase shift ensure a very good quality of the input current drawn from the batteries. As the assumed gain of the DC-DC converter is high (up to 17), the duty ratio varies from zero to 95% at nominal 400 V (Table 1).

**Table 1.** Selected parameters of double-boost DC-DC converter.

| Parameter | Symbol | Value |
|---|---|---|
| Input voltage | $V_{BAT}$ | <24 V |
| Output voltage | $V_{DC}$ | <450 V |
| Switching frequency | $f_s$ | 100 kHz |
| Input inductors | $L_1, L_2$ | 30 µH/30 A max |
| Capacitors | $C_1, C_2$ | 40 µF/250 V $4 \times 2.2$ µF/250 V |
| Duty cycle range | $D$ | 0 ÷ 0.95 |

### 3.2. Isolated Resonant DC-DC Converter

The main aim of the output stage is to boost the output voltage of the DC-DC converter, at nominal conditions from 400 V to 50 kV; moreover, for isolation between the input and output, a high-voltage circuit is provided. As a result of the literature review and analysis carried out earlier, the series resonant converter was selected (see Figure 5a), which, in practice, works with four series-connected secondary windings and sets of high-voltage rectifiers (Figure 5b). Note that the converter is operating at a fixed frequency (65 kHz) and a suitable, also fixed, phase shift while the voltage control is conducted via the input DC-DC converter.

On the basis of simulation analysis, the prototype was developed on a common power board with the DC-DC converter (see Figure 4b). The main circuit of the converter consists of two inverter branches, each of them with two SiC MOSFETs (C3M0065090D) suitable for high-frequency operation. The transistors are controlled by integrated gate drivers (ACPL-P343-000E), which also provide isolation between the control signals and the power circuit. Similar to the input stage, the same DC-DC converters (MGJ2D241505SC) are applied. At the input of each H-bridge, a dry capacitor (40 μF/450 V) is applied but each inverter branch also contains fast capacitors (3 × 1 μF/450 V) supporting fast switching transients (all parameters in Table 2). The resonant tank contains, in addition to the leakage inductance of the transformer, an additional inductor of 24 μH and a set of capacitors with an equivalent capacitance of 9.9 nF.

**Table 2.** Selected parameters of resonant converter.

| Parameter | Symbol | Value |
|---|---|---|
| Input voltage | $V_{DC}$ | 400 V |
| Switching frequency | $f_s$ | 68 kHz |
| Resonant inductor | $L_s$ | 24 μH/35 A max |
| Resonant capacitor | $C_s$ | 3 × 3.3 nF |

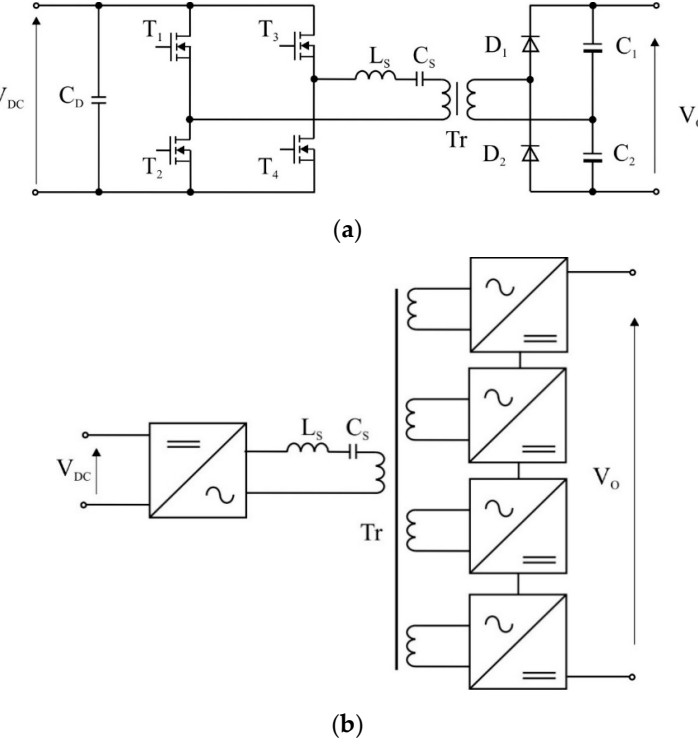

**Figure 5.** Scheme of the single resonant converter module (**a**) and complete schematic with four rectifiers (**b**).

High-voltage rectifiers are an important part of the output stage, operating at voltages up to 50 kV. In this prototype, three half-bridge rectifiers with the neutral point of the capacitive divider were used in series. Due to high-frequency switching and the low current at the high-voltage output, capacitors were 1 nF each, rated at 10 kV. As a rectifier diode, a series connection of 26 SF1600s (1.6 kV/1 A) was applied, which made it possible to block 1/3 of the rated voltage (50 kV) with a necessary margin—the photo of the rectifier is shown in Figure 6. This rectifier is designed to provide air isolation between key points in the circuit. Furthermore, to ensure adequate isolation, the target version was placed in epoxy resin—a view of the housing for 4 rectifiers, made on a 3D printer before pouring, is shown in Figure 6.

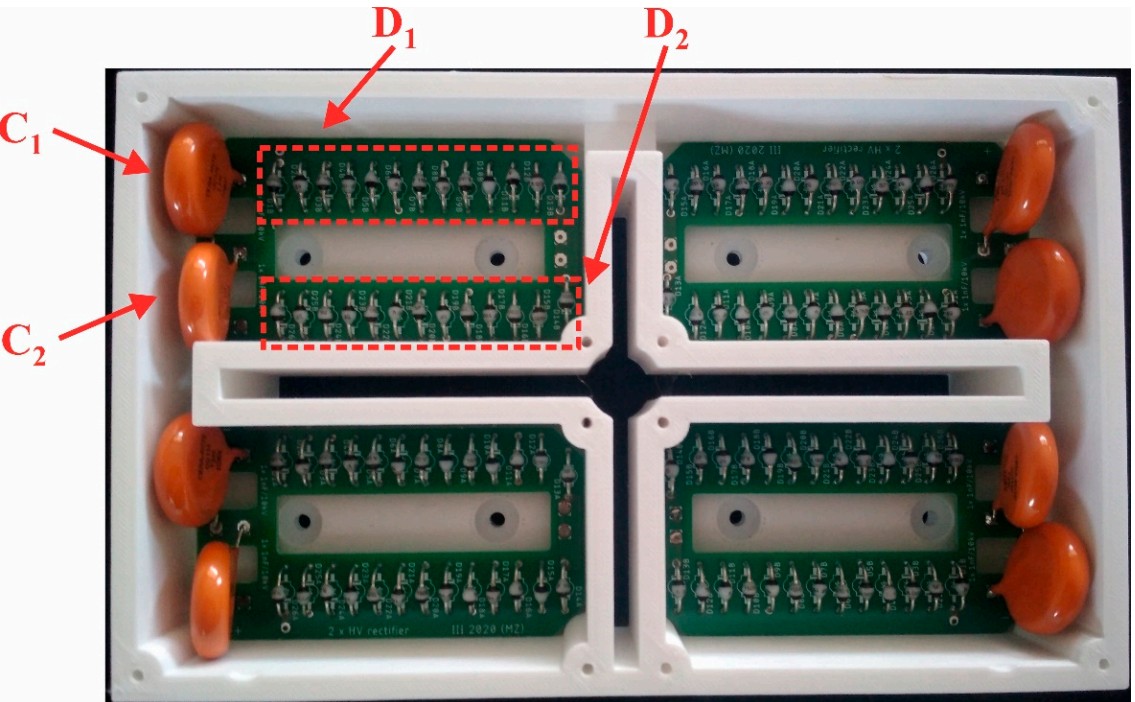

**Figure 6.** Photo of the high-voltage rectifiers inside housing; markings refer to Figure 5a.

### 3.3. High-Voltage Transformer

The applied transformer with a step-up ratio (n = 18) consists of one primary winding (N1 = 6) and four secondary windings (N2 ÷ N5 = 108). The primary winding (N1) is powered by a resonant converter, where the assumed voltage value does not exceed 400 V. With the assumed high-frequency operation (fs = 68 kHz), a Litz wire was used, consisting of 245 insulated wires with a diameter of 0.1 mm each and a total effective cross-section of Scu = 1.92 mm2. Enameling of the individual conductors and the silk braid used were enough to protect the primary winding against breakdown. Secondary winding sectioning allowed the transformer secondary voltage to be divided by 4, so that the maximum voltage on each section is 25 kV ÷ 4 = 6.25 kV. In this case, a single coil with a diameter of 0.25 mm was used, appearing in triple insulation (TIW—triple-insulated wire), which allowed for protection against possible voltage breakdown. Moreover, the prepared windings were separated with a distance of at least 2 mm from each wall of the carcass, and an insulator in the form of a two-component polyurethane cast resin (PUR) was used.

The transformer is equipped with a core, which is a set of 8 fittings (16 halves) of the U 80/49/20 core. The design of the magnetic element was conducted in such a way that the primary winding of the transformer includes 4 fittings with a total core cross-section SFe (N1) = 4 × 400 mm$^2$ = 1600 mm$^2$, while each of the four secondary windings is made on 2 cores with a total cross-sectional area of SFe (N2 ÷ N5) = 2 × 400 mm$^2$ = 800 mm$^2$—Figure 7.

Each of the four secondary windings is connected to a rectifier through specially designed and made connectors with high breakdown strength.

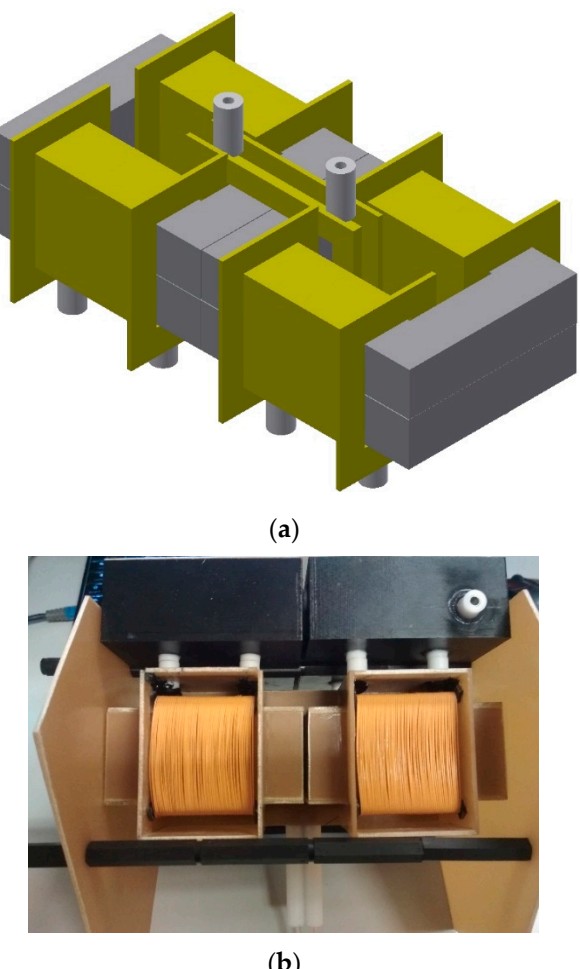

(**a**)

(**b**)

**Figure 7.** 3D visualization (**a**) and photo (**b**) of the transformer.

### 3.4. Digital Control System

The basic assumption of the developed control system is to control two converter systems in such a way as to maintain the value of the output voltage $V_O$ * set by the user without exceeding the rated current at the input (battery) and in all components of the system. It is necessary to ensure cooperation of individual stages: the output, which is the resonant DC-DC converter with the high-frequency transformer, and the input double-boost converter that adjusts the voltage of the $V_{DC}$ intermediate circuit to the appropriately scaled voltage value at the $V_O$ output. In fact, the input stage consists of two voltage boost converters (Figure 4a). Each of them has a separate control system, consisting of an internal current control loop and an external voltage control loop (see Figure 8) with PI controllers. In order to reduce the non-linearity of the characteristic of the boost converter, especially at very high values of the duty cycle, the output signal of the PI controller of the current loop is converted according to

$$D = 1 - \frac{1}{U_P} \tag{1}$$

where $D$ is the duty cycle of PWM pulses, and $U_P$ is the output signal from the current regulator.

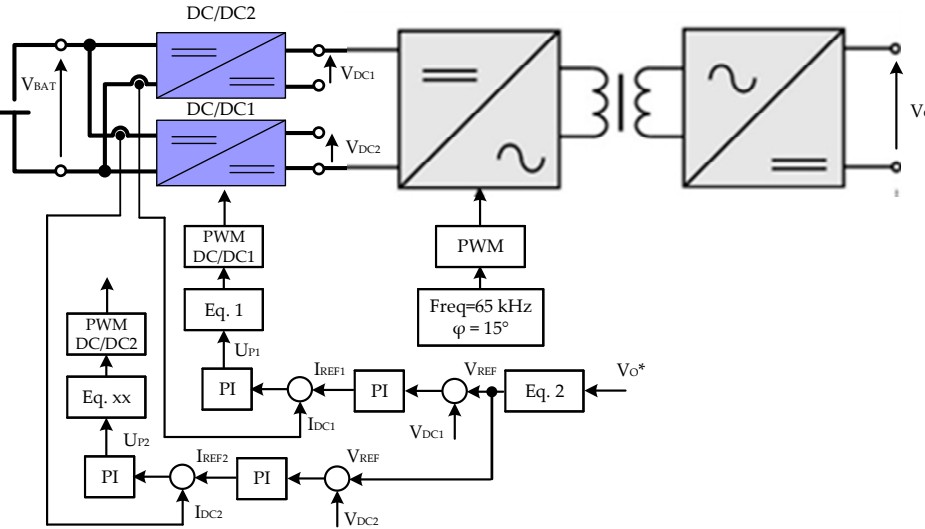

**Figure 8.** Block scheme of the digital control system.

According to the assumed operation principles, the boost converter acts as a voltage source for the resonant DC-DC converter, which is operating at a fixed frequency and phase shift. However, the voltage gain *G* of the resonant converter is not constant and drops with the output voltage/power. Therefore, for the assumed range of the output voltage, the characteristics $V_{REF} = f(V_O)$ were obtained on the basis of experiments and then used to determine the function correcting the set value of the output voltage. The obtained characteristic (Figure 9) was approximated by a third degree polynomial of the following form:

$$V_{REF} = 0.0921V_O^2 + 7.4882V_O - 43.961 \tag{2}$$

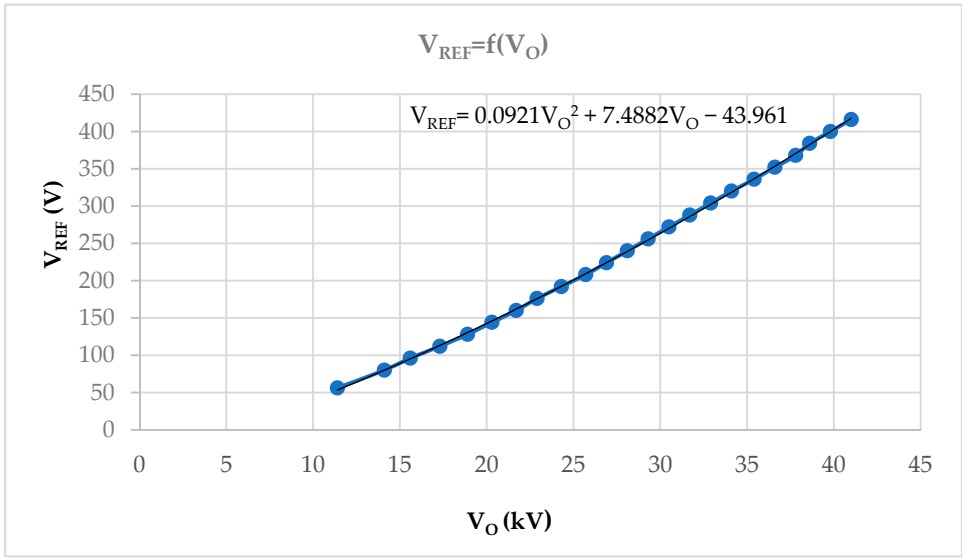

**Figure 9.** Experimentally determined function correcting the relation between input voltage and gain of the resonant DC-DC converter.

The reference voltage $V_O{}^*$ is converted into the appropriate value of the set voltage of the input stage according to Equation (2).

The Texas Instruments TMS320F28377S signal processor (Texas Instruments, Dallas, TX, USA) was used to implement such a developed algorithm—a suitable control board

was developed to control both DC-DC converters and provide a user interface. The photo of the control board is shown in Figure 10.

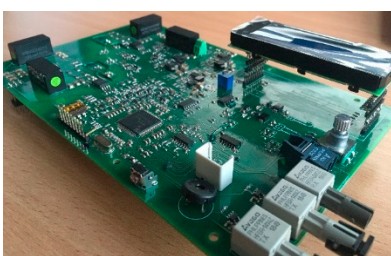

**Figure 10.** Photo of the developed digital control system.

### 3.5. Complete DC Supply

The individual components were combined by keeping the insulation distances and using a glass-epoxy laminate, which is the internal cover of the device. Input terminals (supplying 24 V voltage from the battery) are placed, together with the input fuse, on the left side (see Figure 11). On the other hand, the output of the device (voltage up to 50 kV) is provided in the rear part of the housing, but the positive terminal has been appropriately separated from the metal parts connected to the negative output pole. The front side of the power supply, being the user interface, includes the main switch with the LED indicator (upper left corner), LCD display with a multifunctional selection knob (upper right corner) and a mode switch (MANUAL, REMOTE) with LED indicators (central part of the panel). On the right side of the device, on the other hand, there are fiber optic communication connectors for connecting the superior power supply control system and communication with the device. The presented prototype of the power supply has the dimensions of 242 (width) mm × 298 (length) mm × 136 (height) mm, and thus the volume is slightly below 10 dm$^3$ and the obtained DC power supply can fit into the housing of the Marx generator and may be easily transported.

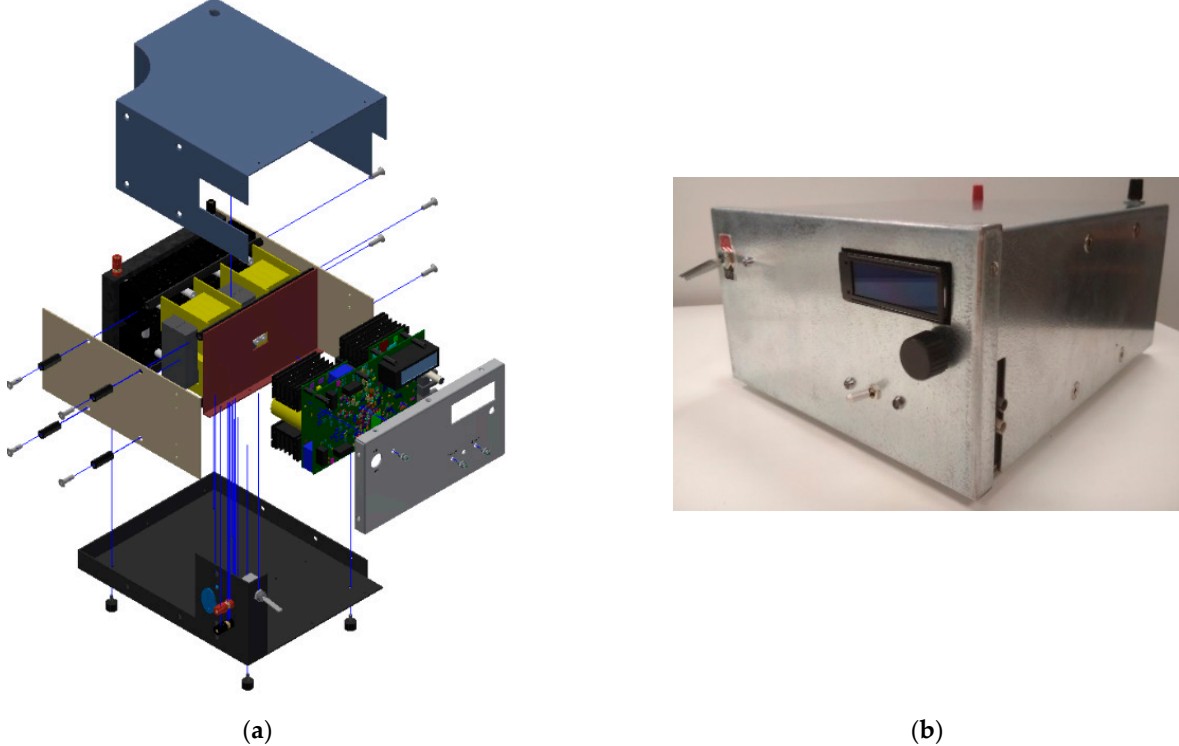

(**a**)                                                                 (**b**)

**Figure 11.** CAD drawing of the developed power supply (**a**) and photo of the finalized prototype (**b**).

## 4. Experiments

After initial tests including the operation of the two DC-DC converters separately and tests of the control system, the system was completed and mounted in the housing (Figure 11). Figure 12 shows the output voltage of the DC power supply without the external load, while the internal load is always the 48 MΩ resistive divider (the part of the $V_O$ measurements) designed to draw 1mA at the nominal output voltage. As can be seen in Figure 12, where the output voltage is presented, the DC supply provides a nominal voltage of 50 kV after 700 ms from the system start. Next, tests were conducted with an adjustable spark gap (see Figure 13) connected via a series resistor, and the DC supply was programmed to keep the voltage for a certain time period. During this time, the spark gap fired at a certain voltage level (5 kV—Figure 13a, and 16.9 kV—Figure 13b), reducing the voltage to zero, and then the supply was charging the voltage to the same critical value. This process was repeated several times, showing very good performance of the supply.

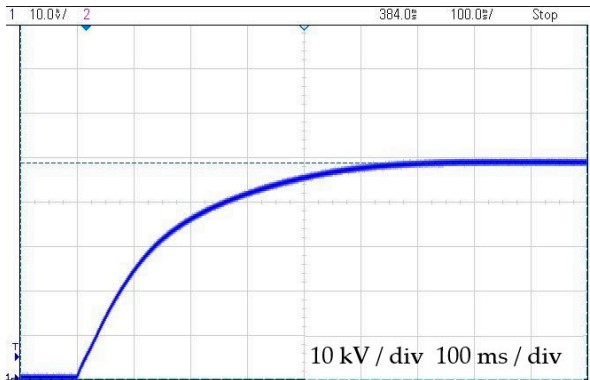

**Figure 12.** Rise of the output voltage from zero to nominal 50 kV without any external load.

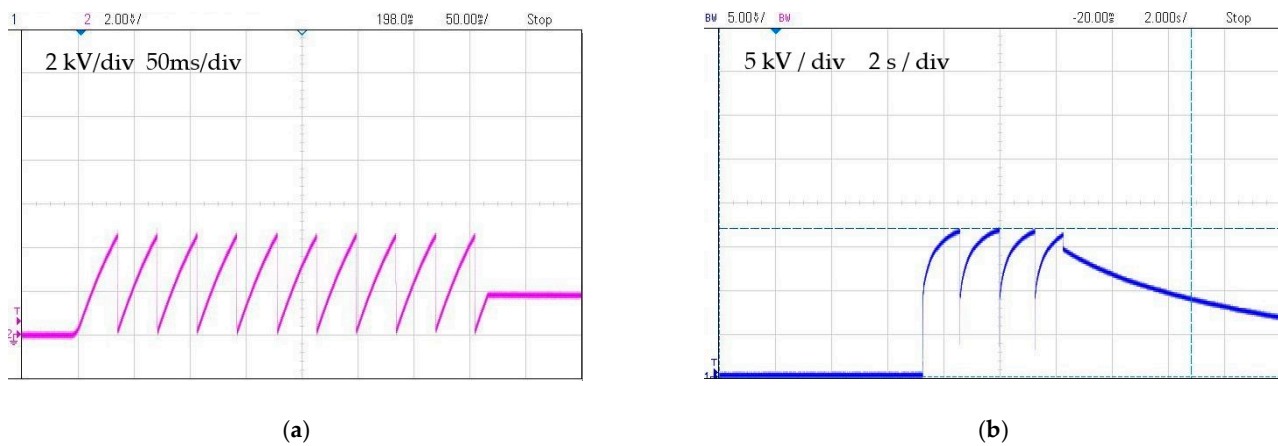

(**a**)   (**b**)

**Figure 13.** Output voltage measured when the spark gap is connected and firing at 5 kV (**a**) and 16.9 kV (**b**).

Next figures illustrate behavior of the DC-DC converters during both tests. The output voltage of the double-boost converter reaches nominal value after 320 ms, however, waveform shows two overshoots due to impact of the current controller (Figure 14). For this operation point a higher proportional gain of PI controller (Figure 8) would help to reach nominal current of 30 A faster but the same settings of the PI-controllers are expected to work properly also at low output voltages. Thus, the introduced settings and presented waveforms are a compromise, acceptable performance is observed for the whole operation range. As the resonant DC-DC converter operates with fixed frequency and phase shift, the peak value of inverter output voltage follows waveform of the $V_{DC}$ (Figure 15). From the same figure can be also seen that current of the resonant tank depends on the conditions in

the inverter and transformer circuit, which is also visible in Figure 16 where waveforms measured after spark-gap discharge are presented. After rapid decrease of the output voltage, the converter is responding by means of increased resonant current and the output voltage is rising again to reach steady-state.

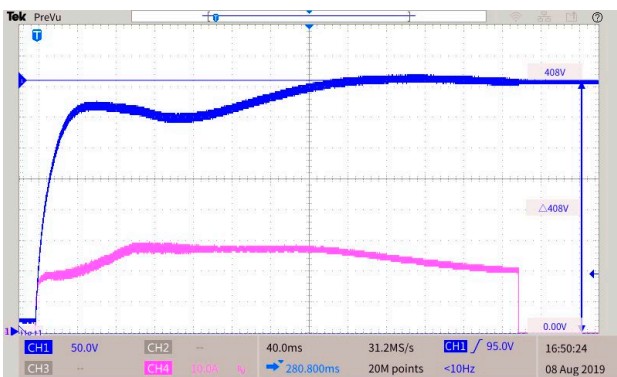

**Figure 14.** Operation of the double-boost DC-DC converter during pulse operation (DC-link voltage $V_{DC}$ and input current).

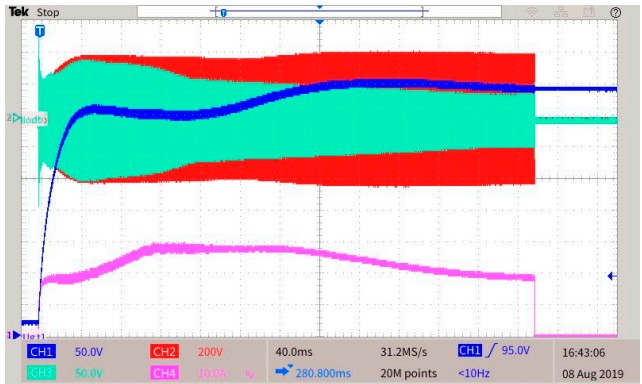

**Figure 15.** Pulse operation of the double-boost converter (DC-link voltage $V_{DC}$ and input current) and resonant converter (inverter output voltage and current of the resonant tank).

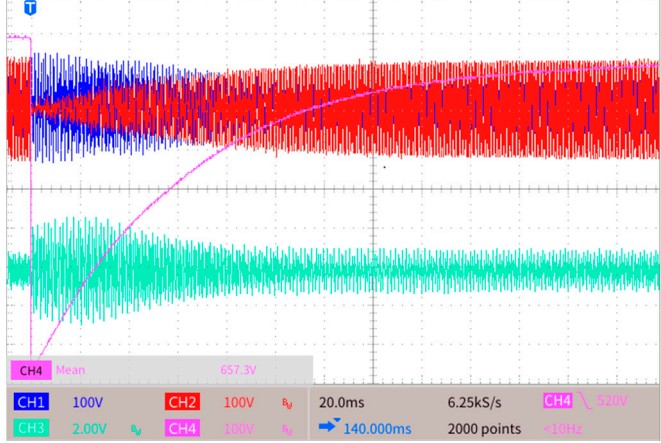

**Figure 16.** Resonant converter operation after output discharge by the spark gap (output voltage, primary voltage of the transformer, resonant capacitor voltage and current of the resonant tank).

Finally, the portable DC supply was connected to the Marx generator in the setup described in Section 2. During the test, the generator was providing pulses counted in

hundreds of kVs, resulting in strong electromagnetic fields emitted by the antenna. Thus, the whole device was controlled via fiber optic links, but the use of an oscilloscope and high-voltage probes near to the test setup was too risky. The performance of the DC supply was watched indirectly by recording voltage measurements in the microprocessor memory—the system was providing the requested voltage to the generator.

## 5. Conclusions

A portable DC supply was designed, built and experimentally validated under various scenarios and circuit conditions, including the Marx generator. Providing voltages up to 50 kV from a 24 V battery was not a trivial task but the applied two-stage solution with two DC-DC converters containing fast-switching SiC power devices proved to be the correct one. The high switching frequency up to 100 kHz enabled a system size reduction ($< 10$ dm$^3$), especially for the inductors and transformer. This component as well as the high-voltage rectifiers was truly demanding due to isolation requirements. According to the presented results, all components were operating correctly in terms of electrical and thermal performances. The proposed control method with the input DC-DC operating as a controllable voltage source and isolated DC-DC converter working at a fixed frequency and phase shift seems to also be suitable for the task. Waveforms presented for the charging or recharging after rapid discharge of the output voltage confirm that the system is stable and controllable. Finally, the user interface and fiber optic links make cooperation with the Marx generator or other loads rather simple.

**Author Contributions:** Conceptualization and methodology, J.R. and J.S.; validation, M.Z., G.W. and A.Ł.; writing—original draft preparation, J.R.; writing—review and editing, A.Ł., M.Z. and G.W.; funding acquisition, J.S. All authors have read and agreed to the published version of the manuscript.

**Funding:** This research received no external funding.

**Data Availability Statement:** The data presented in this study are available in this paper.

**Conflicts of Interest:** The authors declare no conflict of interest.

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
