# Peer review of "Portable DC Supply Based on SiC Power Devices for High-Voltage Marx Generator"

_electronics, doi:10.3390/electronics10030313_

Round 1

Reviewer 1 Report

The paper proposed by the authors to be published in the Electronics journal follows all the rules of the academic journals. The paper is also in agreement with the specific rules of the research topic.

Figures 1 is lacking clarity. It must be sharper as image by improving the resolution or redesign. This figure is the global vision of article concept and must be improved. Figure 3 contains non-English information and some troubles with alignment, probably generated by pdf translation.  Some figures based on oscilloscope captures (13, 16) are lacking in clarity or resolution; if it is possible the be upgraded, it is recommended too. Generally, in order to make the article more accessible for all readers of the Electronic journal, the abbreviations must be explained at first usage, even if they are accepted and standardized in the high voltage power supply topic. Unexplained abbreviation can be a source of confusion too, for example:

(42)  … SiC…  can be … Silicon Carbide (SiC)… where SiC is MOSFET technologies

(128) … LEM current sensors …  can be … Life Energy Motion (LEM®) …  where LEM is the name of the Company that produce the sensor type LA 25-NP

The sections: Author Contributions, Funding, Institutional Review Board Statement, Informed Consent Statement, Data Availability Statement, Conflicts of Interest are uncompleted in this version of the manuscript.

Author Response

Dear Editors and Reviewers,

thank you for your valuable comments to our paper. We have introduced a series of improvements according to all recommendations – all changes are marked with a yellow background.

Reviewer 1

  1. Figures 1a, 1b, 4b, 6, 12, 13 have been improved.
  2. We have introduced more literature references new [3-5], [9],[11].
  3. Explanations of abbreviations have beed added in lines 41, 135.

The authors

Reviewer 2 Report

The manuscript describes major issues related to a design of a portable SiC-based DC supply developed for evaluation of a high-voltage Marx generator. In general, the manuscript is quite interesting to be accepted for publication.

However, some important items to be coonsidered are the following:

1.- In Fig. 3, please, improve the block diagram, including the translation of terms from Polish into English.

2.- In Section 2 (A setup of the Marx generator), or 3 (Portable DC power supply), please, clarify and discuss the final application or equipment of the prototype implemented by authors.

3.- Notice that Section "Author Contributions" should be completed.

Author Response

Dear Editors and Reviewers,

thank you for your valuable comments to our paper. We have introduced a series of improvements according to all recommendations – all changes are marked with a yellow background.

Reviewer 2

  1. Fig 3. has been improved.
  2. In our text application is given: „to generate nanosecond high-voltage pulses for exposure tests of electronic equipment”.
  3. Author contributions have been completed.

The authors

Reviewer 3 Report

The manuscript “Portable DC supply based on SiC power devices for high-voltage Marx generator” is well-written and data is well-presented. Also, this article deals with a topic that fits well in the scope of the journal. I recommend the article for publication in Electronics. However, the following suggestions could be taken into account to improve the manuscript quality:

  • In the Introduction section, please include more of the relevant literature.
  • Please Improve the quality of some of the figures (e.g. Fig 1-b, Fig. 4-b,…) and use labels for the components in Figures (e.g. Fig 4b, Fig 6,…).

Author Response

Dear Editors and Reviewers,

thank you for your valuable comments to our paper. We have introduced a series of improvements according to all recommendations – all changes are marked with a yellow background.

  1. Five new references have been added : [3-5], [9],[11]).
  2. Figures 1a, 1b, 4b, 6, 12, 13 have been improved.

The authors